# A lexicon-based diachronic comparison of emotions and sentiments in literary translation: A case study of five Chinese versions of *David Copperfield*

**Yan Li**[1,2], **Muhammad Afzaal**[3]*, **Yixin Yin**[4]

**1** College of Foreign Languages, Zhejiang Normal University, Jinhua City, China, **2** Graduate School of Translation and Interpretation, Beijing International Studies University, Beijing City, China, **3** Institute of Corpus Studies and Applications, Shanghai International Studies University, China, **4** Press Institute, Beijing Institution of Graphic Communication, Beijing City, China

* muhammad.afzaal1185@gmail.com; Afzaal@shisu.edu.cn

**Data Availability Statement:** Yes - All relevant data are within the manuscript and supporting information files.

**Funding:** This work was supported by Zhejiang Women's Federation Foundation (Grant#202214).

## Abstract

In the Chinese context, translations have served as a useful conduit for providing access to wider literature authored in other languages. A prominent question has been whether translators' linguistic choices are influenced by factors such as translators' social and cultural background and emotions towards the texts they are translating. When multiple translations of the same text over a span of time are produced, another layer of complexity is introduced, and research such as the present study, must examine how or whether variation in the expression of emotions within translations produced over a period of time is discernible. To this end, the present study made use of Lexicon-based Sentiment Analysis (LBSA), a common natural language processing (NLP) approach, to study people's attitudes, opinions or emotions towards a certain person or thing. LBSA has attracted much attention in the literary works or translated works for analyzing reader response and appraisal of the works themselves. The present study undertook a diachronic comparison of emotions and sentiments in five translations of David Copperfield based on the emotion lexicons. The corpus of the study comprised translations of five books and 3,084,599 tokens. We applied the computational method of emotion and sentiment analysis to the emotion words in the five translations. In addition, we used python and R package to analyze the positive and negative words in five versions. The study revealed that translators as social beings in the target world express unique reactions towards the same emotion in the original text as well as in literary translations. Yet, the modern vernacular Chinese versions also showcase a similarity in the expression of emotions thus demonstrating the decisive role of the overall flow of emotion in the original plays and in translation. The contribution of the study is significant as it is a pioneering investigation given that it undertakes a sentiment and emotion analysis of literary translations in Chinese.

**Competing interests:** NO

## Introduction

The rapid development of artificial intelligence (AI) has led to the upsurge of natural language processing (NLP) in a variety of disciplines and scientific fields, which phenomenon has given birth to a new field of research in which human emotions may be traced, perceived and understood by AI [1]. For this reason, sentiment analysis which was originally used to analyze public mood and views also became known for opinion mining and emotion recognition. With the combination of different disciplines ranging from computer, cognitive and social sciences and psychology, it has been developed to extract people's thoughts, feelings and emotions within literary contexts [2], thus making it possible to adopt sentiment and emotion analysis in literature and literary translation studies. Although computer science sentiment analysis is different from emotion identification, they are both subsumed under the Affective Computing umbrella [1], in which the AI researchers show a great interest in human emotion, since their nature and innate ambiguity may prevent them from proceeding human emotions into structured data [3].

At present, two main approaches have been developed to deal with issues on emotions and sentiments which include lexicon-based and machine-learning-based approaches [4]. The latter requires researchers to empower machines with human cognitive capacity so that they can recognize, react and express emotions and sentiments like humans. To achieve this, machines need to undertake deep learning to understand the complex syntactic and semantic norms in a language [1,5,6] requiring a sea of data for machine learning so that machines might identify the language characteristics from the huge number of texts. This is computationally heavier than the lexicon-based approach. On the other hand, lexicon-based approach has been proved to be accurate in analyzing sentiments and emotions in texts, demonstrating an overall accuracy score of 78% [3]. Lexicon-based approach has been adopted in a number of studies on selected languages, ranging from text-based emotion detection for French dataset [7], sentiment analysis of Urdu blogs [8], automatic expansion of Arabic sentiment lexicon through deployment of word embeddings [9] and sentiment analysis of Bangla language using an extended lexicon data dictionary [10] to sentiment analysis of Saudi dialect tweets [11]. In China, this approach has mainly been used to analyze the reader acceptance and appraisal of literary works or literature translation [12,13]. Yet little research has been made to analyze emotions and sentiments in Chinese translated literary works.

Against this backdrop, the present study focuses on the lexicon-based sentiment and emotion analysis in translated texts into English from Chinese. Five Chinese translations of David Copperfield by Lin Shu (1852–1924) (Lin), Xu Tianhong (1907–1958) (Xu), Dong Qiusi (1899–1969) (Dong), Zhang Guruo (1903–1994) (Zhang) and Zhuang Yichuan (1933-) (Zhuang) were selected. The translations were respectively diachronically published in 1908, 1943, 1950, 1980 and 2000. The reasons for the selection of these five versions was due to their correspondence with particular eras in Chinese history. According to the publishing time, Lin's version was published in late Qing Dynasty (1636–1912), Xu's in the Republic of China (1912–1949), Dong's in the beginning of the founding of People's Republic of China (PRC, Oct. 1949-), Zhang's at the dawn of Reform and Opening-up policy process in PRC (which was first implemented in 1978), and Zhuang's at the start of the new millennium. Spanning over a century, the five versions may provide insights into the changes in the presentation of emotion and sentiment presentation over the period of one-hundred years, thus helping to probe the epochal characters of emotion and sentiment expressions in each version.

When it comes to the language, Lin's version was written in ancient Chinese, whereas the remaining four were rendered in modern vernacular Chinese. Although the first four translators were all born in the late years of Qing Dynasty, Lin translated this book before the modern vernacular Chinese and the New Culture Movement at the time of the May 4th Movement in

1919, while others wrote the translations later than him. After the establishment of new China on October 1, 1949, the modern vernacular Chinese became the common written language in China. Xu Tianhong did not choose ancient Chinese as he did not believe a comprehensive and real Dickens could have been shown to Chinese readers in Lin's version. Only by translating in modern vernacular Chinese could this goal be achieved [14]. In that case he also chose modern vernacular Chinese. Thus, the analysis of the emotion words of Chinese in different stages may exhibit the development of Chinese from ancient to modern times.

Further, the five versions were widely accepted among the Chinese people. Lin's 1908 translation of David Copperfield proved so popular with readers that six different editions were released from 1908 through 1925. It is still regarded as a classic. Xu's translation as the first modern vernacular Chinese versionwas also published in Taiwan. Dong's version has also proved popular amongst Chinese readership from 1950s to 1970s, and five editions were released during that period. Zhang's version was once taken as the best translation of the novel since its publication [15].

It is therefore evident that the five versions are all in line with the public aesthetic expectations of their time and the emotions contained in the translations resonate with the readers, thus complying with the mainstream ideology. In such a case, a diachronic comparison of emotions in these translations can make it clear how differently translators deal with emotions and what factors may contribute to their choices. Based on the findings, the common norms as well as the unique features of emotional expressions in translated literature may be identified.

### Research questions

In accordance with the above purpose, the study is going to explore the answers to the following questions:

I. To what extent is it possible that the dictionary-based sentiment analysis can be applied to Chinese translated works published at different times?

II. What is the degree to which emotions can be extracted from Chinese translations of literary works carried out at different times?

III. What factors may contribute to the extraction of emotions of Chinese translated literature?

## Materials and methods

The study adopted the computational method of emotion and sentiment analysis to extract the emotion words in the five selected translations. In dealing with emotions and sentiments in English literature, scholars have designed and applied different packages such as *syuzhet* and *sentimentr* used in R, a free software environment for statistical computing and graphics. As *syuzhet* package was originally designed by Matthew Jockers for accessing the computational plot and sentiment extraction. However, it did not take into consideration the influence of valence shifters such as negators (not), intensifiers (highly), downtoners (slightly), or adversative conjunctions (however) each of which has the potential to change the sentiment values of the sentence. On the other hand, *sentimentr* solves this problem by integrating valence shifters into computational analysis in order to more accurately calculate text polarity sentiment in the English language at the sentence level through optional aggregation by rows or grouping of variable(s).

However, these packages are not suitable for sentiment analysis of the Chinese language. Although they are equipped with the NRC emotion intensity lexicon which has been google

translated into more than a hundred languages including Chinese, there are several problems in the translation.

In NRC, English emotion intensity lexicon emotions are divided into eight types, and the total number of emotion words amount to 9921, as is shown in Table 1 below:

Besides these words, there are 4261 words with zero emotion intensity in the lexicon. Altogether 14182 words are included in the NRC-EIL. Yet while translating these words into Chinese, the same Chinese translations are available. For example, "penny" and "cent" are both translated into "一分钱", "morsel" "mouthful" into "一口", "coincident" "unanimity" "congruence" "unanimous" and "unanimously" all into "一致". In such a case, it is difficult for the machine to decide which English word with its emotion intensity should match the Chinese word.

What is more problematic is the inaccuracy of translation. In translating the English words, only one Chinese word is chosen as the corresponding translation by Google Translate (GT). However, one English word may have more than one meaning in Chinese, or the Chinese translation may have its own meaning in the Chinese environment that diverges from the meaning of the English word. "Band" is translated as "带(*dai*)", which can be both a noun and a verb. When used as a noun, the character "带" usually is not used alone, but attached by a modifier, like "子(*zi*)", and the combined characters "带子(*dai-zi*)" means "band". Without *zi*, *dai* is often regarded as a verb, meaning take, bring, carry or even have in English. Thus, both the meaning and part of speech undergo change.

Another difference between English and Chinese is that while one English content word may have a definite meaning, a single Chinese character is unlikely to do so. For instance, "love" is "爱(*ai*)" in Chinese, but *ai* can also combine with other characters and have a different meaning other than "love". In this novel a girl named Emily has been translated as "爱密柳(*ai mi liu*)" by Lin Shu, "爱米雷(*ai mi lei*)" by Xu Tianhong, and "爱弥丽(*ai mi li*)" by Dong Qiusi and Zhang Guruo. All three characters together offer the transliteration of the English name. However, when the NRC-EIL translation dictionary is used, "*ai*" is segmented from the other two characters and has the meaning "love", which is evidently not correct.

Therefore, NRC Chinese dictionary was not viewed as being an appropriate choice for use in the present study.

At present, quite a few Chinese emotion lexicons have been developed, such as HowNet by China National Knowledge Internet, NTUSD by Taiwan University, LI Jun's Positive and Negative Chinese dictionary and BosonNLP. However, these dictionaries are either directed towards commercial use or integrated with a limited vocabulary which is inadequate for meeting the needs of the abundant emotion words in literature. Through careful comparison, the DLUT-Emotionontology by Dalian University of Technology with more than 20,000 emotion

**Table 1. Emotion categories and Word numbers in NRC Emotion Intensity Lexicon (NRC-EIL).**

| Emotion categories | Word numbers | percentage |
|---|---|---|
| anger | 1483 | 14.95% |
| anticipation | 864 | 8.71% |
| disgust | 1094 | 11.03% |
| fear | 1765 | 17.79% |
| joy | 1268 | 12.78% |
| sadness | 1298 | 13.08% |
| surprise | 585 | 5.90% |
| trust | 1564 | 15.76% |
| total | 9921 | 100% |

words reflecting five-range intensity and positive or negative sentiment was chosen for analyzing the emotions and sentiments in the Chinese versions. All those words are divided into 7 types of emotions, namely, 好(*hao*, goodness), 乐(*le*, joy), 怒(*nu*, anger), 哀(*ai*, sadness), 惧(*ju*, fear), 恶(*wu*, disgust) and 惊(*jing*, surprise), with altogether 21 sub-types. However, in one of the sub-categories named "思 (*si*, missing)", which is categorized as a positive group with the type expression of PF, there are a number of words and phrases that do not belong to the positive group, like 忧惧(*you-ju*, worry and apprehension), 挂虑(*gua-lv*, worry). Thus we added one more type, named 盼(*pan*, anticipation) with the old type expression of *si* as PF which included all the positive emotions words in *si*. The rest negative emotion words in *si* were kept, with a new subtype expression as NF. The categories, sub-categories, and expressions are presented in Table 2.

What also needs to be noted is that Chinese characters can change their places in a word without changing the meaning, and there are many expressions of the same meaning, like "见多识广 (*jian-duo shi-guang*)" "识多见广(*shi-duo jian-guang*)", both meaning "experienced and knowledgeable", and "三心二意 (*san-xin er-yi*)" "三心两意 (*san-xin liang-yi*)", referring to "in two minds" in which *er* means the same as *liang*. In view of this, a Chinese thesaurus (http://github.com/guotong1988/chinese_dictionary) was also added to the analysis. There is also a list of stop words. In Chinese, "先生（*xian sheng*)" can be a neutral word, not only used as a prefix to a male surname but also used as a positive word for showing respect to a teacher or a master. In the novel, this mainly refers to the first meaning, whereas in the Chinese lexicon, it is regarded as a positive word with a high intensity. Therefore, this word is listed in the Stop Words. In the lexicon, emotion words do not include the adverbs which can change the degree of emotion intensity. A summary of those adverbs was made, establishing an adverb list and giving each adverb an intensity valence according to the degree it modified the emotion word.

In sum, the Chinese lexicon, the Chinese thesaurus and the adverbs and the stop word lists were all adopted to analyze the emotion and sentiment of the translations in the Python environment.

## Results and discussion

### General emotion-word types and tokens

By adopting Python language with the above-mentioned lexicons and word lists, emotion words were extracted from the translations. The positive and negative emotion word types and tokens are listed in Table 3.

From the table above, it is evident that the incidence of emotion words in Lin's version is limited to a great extent in comparison with the other four translations. The reason is simple. Lin Shu translated the text into ancient Chinese. The biggest difference between ancient Chinese and modern vernacular Chinese is that whereas in the former single characters have concrete meaning, in the latter, often two or more characters combine together to express a certain meaning. For example, "吾夫妇之居此，乐至矣。(*wu fu-fu zhi ju-ci, le zhi yi*)"in Lin's version (original: . . .Mrs. Micawber and himself had been made extremely snug and comfortable there), if translated into modern vernacular Chinese, the sentence might be "我们夫妇 居住在这里，快乐至极呀 (*wo-men fu-fu ju-zhu zai zhe-li, kuai-le zhi-ji ya*)" (We couple live here and feel extremely joyful). *le zhi* may mean "extremely happy", or "joy comes", for *zhi* may be an adverb of the strongest degree or a verb meaning come or reach. Yet *le* can't be extracted as an emotion word, as words in the Chinese emotion lexicon are mainly modern vernacular Chinese, and few single Chinese characters are listed in it. As such, only those words that are still used in modern Chinese can be extracted. Therefore, it is safe to conclude

**Table 2. Emotion category.**

| Serial number | Emotion category | Emotion sub-category | Sub-category expression | Example |
|---|---|---|---|---|
| 1 | 乐(*le*, joy) | 快乐 (*kuai-le*) | PA | joy, cheerful, smiling |
| 2 | | 安心 (*an-xin*) | PE | set one's mind at rest |
| 3 | 好(*hao*, goodness) | 尊敬 (*zun-jing*) | PD | respectful, reverent |
| 4 | | 赞扬 (*zan-yang*) | PH | excellent, handsome |
| 5 | | 相信 (*xiang-xin*) | PG | believe, trust, dependent |
| 6 | | 喜爱 (*xi-ai*) | PB | adore, admiration |
| 7 | | 祝愿 (*zhu-yuan*) | PK | blessing, longevity |
| 8 | 怒(*nu*, anger) | 愤怒 (*fen-nu*) | NA | angry, annoy, furious |
| 9 | 哀(*ai*, sadness) | 悲伤 (*bei-shang*) | NB | sad, grief, painful |
| 10 | | 失望 (*shi-wang*) | NJ | despair, frustration |
| 11 | | 疚 (*jiu*) | NH | regretful, guilty |
| 12 | | 思 (*si*) | NF | worry, apprehension |
| 13 | 惧(*ju*, fear) | 慌 (*huang*) | NI | at a loss, flustered |
| 14 | | 恐惧 (*kong-ju*) | NC | timid, horrify, nervous |
| 15 | | 羞 (*xiu*) | NG | ashamed, embarrassed |
| 16 | 恶(*wu*, disgust) | 烦闷 (*fan-men*) | NE | distracted, oppressed |
| 17 | | 憎恶 (*zeng-wu*) | ND | abhor, detest, hatred |
| 18 | | 贬责 (*bian-ze*) | NN | vanity, peacockery |
| 19 | | 妒忌 (*du-ji*) | NK | jealous, envious |
| 20 | | 怀疑 (*huai-yi*) | NL | suspicious, dubious |
| 21 | 惊(*jing*, surprise) | 惊奇 (*jing-qi*) | PC | marvel, amaze |
| 22 | 盼(*pan*, anticipation) | 期待 (*qi-dai*) | PF | aspiration, anticipation |

that only part of emotion words in Lin's version are calculated in this analysis. In this regard, it can easily be gauged from the table that the emotion word types in Lin's version make up more than half of those found in the versions by Xu, Dong and Zhuang and more than 40% of those found in the version by Zhang. This shows that quite a lot of emotion words in ancient Chinese are still in use in modern vernacular Chinese.

In consideration of emotion-word type-token ratio, Lin clearly did not use as many repeated emotion words as the other four. One of the reasons might be that not all emotion words have been drawn from the translation by Lin. It might also be true that Lin Shu has more various ways of expressing emotions than the others which may reflect emotions in the original work as "decisive constraints" that shape Lin's emotion flow in his translation [16]. To

**Table 3. Emotion word types and tokens in five translations.**

| | Lin's | Xu's | Dong's | Zhang's | Zhuang's |
|---|---|---|---|---|---|
| Total Chinese characters | 304,455 | 648,540 | 649,188 | 811,569 | 670,847 |
| Positive emotion word type | 891 | 1513 | 1374 | 1928 | 1525 |
| Positive emotion word token | 2270 | 13191 | 12908 | 13185 | 12261 |
| Negative emotion word type | 780 | 1585 | 1431 | 2020 | 1605 |
| Negative emotion word token | 1836 | 8843 | 8093 | 8954 | 8472 |
| Total emotion word type | 1673 | 3098 | 2805 | 3948 | 3130 |
| Total emotion word token | 4106 | 22034 | 21001 | 22139 | 20733 |
| Token-character ratio | 1.35% | 3.40% | 3.23% | 2.73% | 3.09% |
| Type-token ratio | 40.75% | 14.06% | 13.36% | 17.83% | 15.10% |

expand, Lin wrote in the prescript of his translation for the latter half of the original that his norm of translating the book was that he worked as a puppet of the author in the expression of emotion, observing that when sadness was shown in the narration, he as a reader could not help feeling upset and when reading the lines of joy, his sense of delight also arose from inner heart. The characters in literary works with specific emotions may affect the readers' attitudes and perceptions on people of this kind [17] which may lead to the blending of the emotions of the author and those of the translator in the translated text. Dong's version, on the other hand, has emotions in the least rich and colorful mode. Dong was greatly influenced by Lu Xun, a pioneer of modern Chinese literature and translator who also advocated the notion of literature for life reality rather than for literature itself. Therefore, literal translation-even rigid ones- became their first choice. Dong set ultimate faith as his norm in translating literature, emphasizing the instrumental over the emotional function of literature. To him, form and content weighted heavier than emotions. Therefore, he was likely to neglect the fact that David was Dickens' "favorite son" (Dickens' own words in the preface of the 1867 edition of the book), and choose more neutral words in the translation, thus reflecting a lack of diversity of emotion emergence.

In consideration of both word type and token, except for the version by Lin, all the other four translations have fewer positive emotion word types than negative ones and more positive emotion word tokens than negative ones. It has been noted that emotion words extracted in Lin's version are not complete. Therefore, it might be the case that the other four versions may provide a fully picture of how emotions flow in the novel. Although negative emotion categories exceed the positive ones, the novel, in general, reflects a positive attitude as more positive than negative words have been adopted in it. It conforms to the author's intention of creating the novel. While criticizing the selfishness, coldness, hypocrisy, cruelty and darkness of industrial capitalist society in the 19th century, represented by Mr. and Mrs. Murdstone, Mr. Creakle the headmaster of Salem House, Uriah Heep and his mother, Littimer and Steerforth, Dickens also depicts the goodness, kindness and faith of a group of people, including Aunt Betsey, Agnes, and the poor people like Mr. Peggotty and his family, and even Dick, a man with mental disability. David has experienced the ugliest nature of human beings before he reaches his aunt's house. Yet his life fortune continues to rise under his aunt's guardianship. He finally achieves success in his career, marries Agnes, and establishes a happy family and becomes a renowned author. Charles Dickens may have devoted all his energy and love to David Copperfield, who, having gone through all difficulties, managed to keep his heart pure as gold, eventually achieving great success. This proves Dickens's unique feature of skillfully merging romanticism and realism in a harmonious way.

## Word type and token of each emotion category in the five translations

In the Python environment, all the emotion words were extracted and categorized into 8 emotion categories (Table 2). In each type, there is more than one sub-type which might be of positive sentiment signaling as P, or negative sentiment as N. Altogether, there are nine positive and 13 negative sub-types in the dictionary. Words and phrases of different emotions were extracted from the five translations and the results are listed separately in Tables 4 and 5.

In Fig 1, a comparison of the emotion categories of five translations shows that joy is the main current of all emotions which is followed by disgust. As such, it is evident that five translators have most types of joy in their own versions, although they use different words to describe joy. The same is true for the emotion of disgust of which the types are quite close to those of joy. To a certain extent, this reflects the common norm of translating emotions in literature. In the target environment, different translators may have their own understanding

**Table 4. Comparison of Xu's emotion words in Martin's system and the Chinese dictionary.**

| | | | | | |
|---|---|---|---|---|---|
| Xu's | positive | happiness | 喜欢(+5) | 191 | 好 (goodness) |
| | | | 高兴(+5) | 201 | 乐 (joy) |
| | | | 快乐(+5) | 192 | |
| | | security | 相信(+7) | 300 | 好 (goodness) |
| | | | 一定(+5) | 289 | |
| | | satisfaction | 可爱(+5) | 113 | 好 (goodness) |
| | | | 神气(+5) | 123 | |
| | | inclination | 希望(+5) | 352 | 好 (goodness) |
| | | | 愿意(+1) | 200 | |
| | negative | unhappiness | 难过(-5) | 112 | 哀 (sadness) |
| | | | 痛苦(-7) | 71 | |
| | | | 不幸(-9) | 101 | |
| | | | 悲哀(-5) | 70 | |
| | | | 死 (-7) | 106 | |
| | | insecurity | 怀疑(-9) | 93 | 恶 (disgust) |
| | | | 惊异(-5) | 92 | 惧 (fear) |
| | | | 不安(-5) | 76 | |
| | | | 死 (−7) | 106 | |
| | | dissatisfaction | 卑微(-1) | 66 | 恶 (disgust) |
| | | | 死 (-7) | 106 | |
| | | disinclination | 害怕(-3) | 76 | 惧 (fear) |
| | | | 厉害(-3) | 67 | |
| | | | 死 (-7) | 106 | |

and experiences of emotions which might result in their unique choices of emotion words. Therefore, in general, they abide by the general display of emotions that conform to the target language and culture. It is also shown that emotion categories in a certain society remain comparatively consistent regardless of the passing time. Therefore, a society's ideology of emotions shapes translators' choice of emotion categories which contributes little to the development of language itself. For instance, Lin's version also displays a similar tendency of emotion expressions when compared to the other texts.

In consideration of the emotion word type-token ratio, the emotion words in each category, the modern Chinese emotion dictionary may work, to a certain degree, on the emotion and sentiment analysis of the translated ancient Chinese literature. In approximate calculation, some common norms can be drawn from the results.

In the study, all translators of different times spontaneously adopted similar translation behavior. In their texts, the eight emotion categories successively occupy a similar percentage in the whole emotion flow, with joy and disgust-the two contradictory emotion categories- as the main emotion currents and with surprise and anticipation as the narrowest stream of emotions. Translator behavior is often the result of the joint action of both linguistic and sociological factors [18], which reflects the complexity of influences that may attribute to translators' choice in their translation of emotions. The translators' language aptitude, their inner experiences in reading the original, their social and cultural background and their value and ideology on the world, all may work together in translators' emotional decisions. According to the explanation above, the inheritance of culture in a society is rather continuous and consistent which may be a key reason why translators continue to display the same tendency in selecting emotion categories.

**Table 5. Comparison of Dong's emotion words in Martin's system and the Chinese dictionary.**

| Dong's | | | | | |
|---|---|---|---|---|---|
| | positive | happiness | 喜欢(+5) | 338 | 好 (goodness) |
| | | | 高兴(+5) | 205 | 乐 (joy) |
| | | | 愉快(+5) | 168 | |
| | | | 快活(+7) | 175 | |
| | | security | 相信(+7) | 641 | 好 (goodness) |
| | | | 一定(+5) | 438 | |
| | | | 必须(+7) | 152 | |
| | | satisfaction | 可爱(+5) | 163 | 好 (goodness) |
| | | inclination | 希望(+5) | 362 | 好 (goodness) |
| | negative | unhappiness | 痛苦(-7) | 97 | 哀 (sadness) |
| | | | 不幸(-9) | 102 | |
| | | | 悲哀(-5) | 100 | |
| | | | 忍受(-9) | 72 | 惧 (fear) |
| | | | 死 (-7) | 123 | |
| | | insecurity | 怀疑(-9) | 75 | 恶 (disgust) |
| | | | 吃惊(-5) | 107 | 惧 (fear) |
| | | | 不安(-5) | 107 | |
| | | | 死 (-7) | 123 | |
| | | dissatisfaction | 卑贱(-9) | 77 | 恶 (disgust) |
| | | | 苦恼(-5) | 72 | |
| | | | 陷入(-3) | 66 | |
| | | | 死 (-7) | 123 | |
| | | disinclination | 可怕(-3) | 109 | 惧 (fear) |
| | | | 死 (-7) | 123 | |

It is interesting to see that Lin Shu has the richest way of showing anticipation among the five translators, although his expression of other emotions is more restricted than the other four translators. With anticipation excluded, the other 7 emotion categories appear more in Zhang's version than in the others. One reason for that might be the fact that Zhang has adopted the method of paraphrasing in his translation to make his version sound more smooth, accurate and easy to understand. His translation is the longest one, with 811,569 Chinese characters. In the context of the total emotion word token to Chinese characters ratio, with the exception of Lin's version (since quite a lot of emotion words in it were unable to be drawn) the frequency of emotion words in Zhang's text is less than the other three as shown in Table 3, implying that its emotion density is the lowest among the four modern vernacular Chinese. In this context, the emotion words in Zhang's version are less frequently adopted than in the other three versions. Like waves in a sea may turn weaker if they come one after another with a longer distance, emotions distributed more widely might sacrifice certain amount of intensity in expression. Therefore, it might be safe to conclude that the shorter a piece of writing is, the emergence of emotions is more frequent and the strength of the emotions is also more robust.

When it comes to total word tokens in each emotion as shown in Fig 2), the flow of all categories of emotions in five translations also satisfies the fact that both joy and disgust occupy over two thirds of the whole emotion flow. Yet in contrast to emotion word type within which disgust stands nearest to joy, the ways to express disgust are much closer to the ways to express joy, whereby words of joy are repeated more frequently than disgust. Overall, joy is the dominant emotion in the translations. In consideration of emotion expressions in each translation,

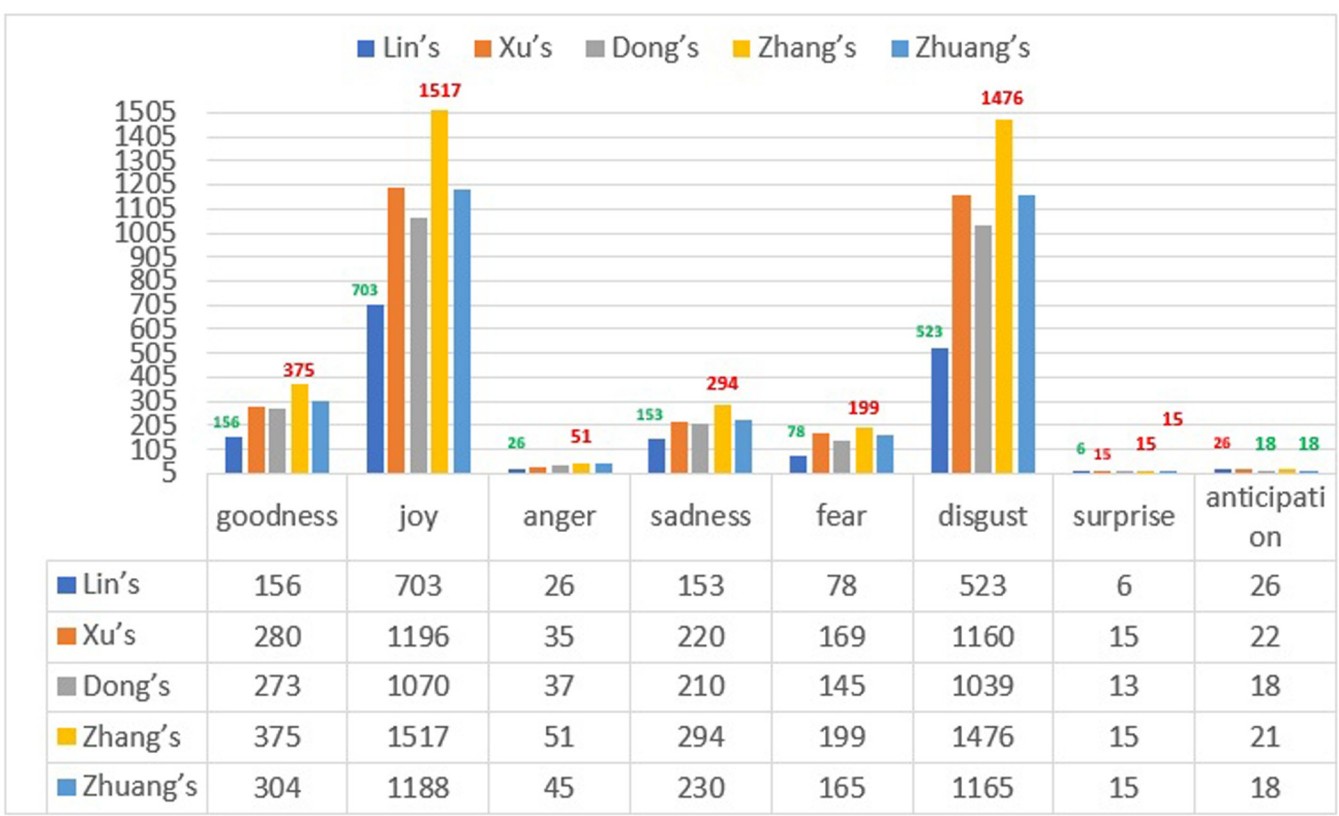

| | goodness | joy | anger | sadness | fear | disgust | surprise | anticipation |
|---|---|---|---|---|---|---|---|---|
| ■ Lin's | 156 | 703 | 26 | 153 | 78 | 523 | 6 | 26 |
| ■ Xu's | 280 | 1196 | 35 | 220 | 169 | 1160 | 15 | 22 |
| ■ Dong's | 273 | 1070 | 37 | 210 | 145 | 1039 | 13 | 18 |
| ■ Zhang's | 375 | 1517 | 51 | 294 | 199 | 1476 | 15 | 21 |
| ■ Zhuang's | 304 | 1188 | 45 | 230 | 165 | 1165 | 15 | 18 |

Word Type count of each emotion type in five translations

**Fig 1. Word type and token of each emotion category.**

it is fairly clear that Zhang's version does not reach the peak of all with the exception of joy, anger and disgust. For goodness, fear and anticipation, Xu Tianhong adopts more words of these emotions than others, while words of sadness in Zhuang's version and surprise in Dong's version are respectively dominant over the texts by others. These results indicate a common tendency, namely that emotions in different translations reflect fidelity to the same original in general flow and remain consistent with the main or the minor currents consistent, although the capacity for either may vary owing to individual and social factors.

Whether consciously or sub-consciously, the translators of the same language have made similar decisions in interpreting the emotions, although they have different types of emotion processing in resulting in their own signature expression of emotions.

Based on the common conformity of emotion types and tokens in five translations presented in Figs 1 and 2, as a general norm of translating emotions in literature, it may be concluded that within a certain culture and society, translators abide by the same rules regulating the categories of emotions.

## Top 10 high-frequency positive and negative words in five versions

Although translators have shown their uniformity in the quantity sequence of emotion categories, they evidence different expressions in each category and the number of times each

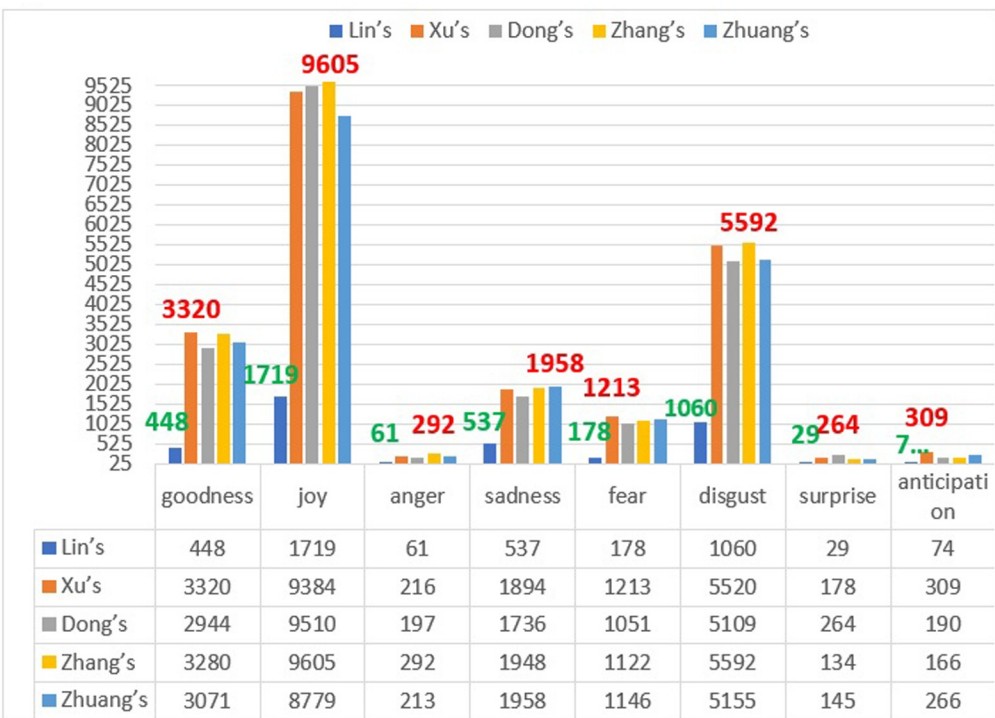

| | goodness | joy | anger | sadness | fear | disgust | surprise | anticipation |
|---|---|---|---|---|---|---|---|---|
| Lin's | 448 | 1719 | 61 | 537 | 178 | 1060 | 29 | 74 |
| Xu's | 3320 | 9384 | 216 | 1894 | 1213 | 5520 | 178 | 309 |
| Dong's | 2944 | 9510 | 197 | 1736 | 1051 | 5109 | 264 | 190 |
| Zhang's | 3280 | 9605 | 292 | 1948 | 1122 | 5592 | 134 | 166 |
| Zhuang's | 3071 | 8779 | 213 | 1958 | 1146 | 5155 | 145 | 266 |

Word token count of each emotion in five translations

**Fig 2. Categories of emotions in five translations.**

emotion word appears is also distinct from one another. It has been discussed that quite a few of the emotion words in Lin's version cannot be drawn as they are single characters in ancient Chinese, like "悦(*yue*, like/love/ joy)" "忧(*you*, worry/anxious/sorry)" "喜(*xi*, happy/love/ like)". Thus, only one single character "死 (*si*, death)" has been extracted. In line with this, there are 891 positive word types, to which there are only 23 frequencies with the total of 2270 times. Similarly for 780 negative ones, there are 21 frequencies occurring 1836 times. Yet for the other four versions, their emotion word types and frequencies are at least twice those found in the version by Lin. Therefore, it might be much clearer to gauge the similarities and differences between the four if more high-frequency emotion words are included. Therefore, the top 30 of both positive and negative ones are listed separately in Appendix I and Appendix II in S1 Appendix for analysis. All Lin's emotion words have been included, but owing to the restrictions of space, it was not possible to list them all herein. Thus, the emotion word list in his version corresponds with those for the texts produced by the other four translators.

On the other hand, only 21 emotion words in Lin's top 14 types with a frequency of over 10 times appear, and some of these have the same frequency. In terms of those words appearing less than 10 times, for each frequency, there are at least 8 or more word-types, and the less the frequency in evidence, the more the likelihood of word types. For this reason, the top 14 high frequently used positive and negative emotion words in the five translations were analyzed first. It is necessary to mention that frequency and word token have the same meaning in this article.

According to the Appendixes, it is evident that different translators have their distinctive choices of emotion expression. Although some words overlap with each other, it is hard to find the same frequency for one emotion word in five translations. Among the top 14 high-

frequency words, only "朋友 (friend)" and "死 (death)" respectively appear as the common positive and negative ones in five versions. The two words exhibit the broadest topic of the translation which may also reflect the general topic of the original, namely how David suffers the death of his parents and little brother, his first wife, his friend Ham and Steerforth, and some less important persons in his life, like Aunt Betsey's and Peggotty's husband. This book also depicts David's friends who have played very important roles in his growing-up, like the knowledgeable and kind-hearted Doctor, the kind and helpful Agnes and Traddles, and even those hypocritical and evil persons like Steerforth and Uriah, optimistic and honest Micawber, poor but brave Ham as well as the mentally troubled Dick.

If all emotion words in Lin's version are taken into consideration, among the top 30 high-frequency words, some more word types can be found. In the positive group, "相信 (believe)" "希望 (hope)" "神气 (manner)" "愿意 (willing)" "精神 (vigor)" "微笑 (smile)" are used in all translations. In the negative one, only four words appear including "不幸 (unfortunate)" "消息 (news)" "眼泪 (tears)" and "困难 (difficult)". "朋友 (friend)" "相信 (trust/believe)" and "希望 (hope)" are all words of joy of which the intensity valence is respectively +9, +7 and +5, and "神气 (manner)" "愿意 (willing)" "精神 (vigor)" "微笑 (smile)" belong to the goodness category with the valence of +5, +1, +7 and +5. "死 (death)" in the dictionary is a "NX" subtype word, which means it may be any "N (negative)" subtype in different contexts. Its valence is -5. "不幸 (unfortunate)" and "眼泪 (tears)" are of high-intensity sadness, as the valence of these is separately -9 and -5. "困难 (difficult)" is a weak sense of fear, with a valence of -1, and "消息 (news)" is a kind of slight worry (-1). Overall, these words exhibit the high-intensity emotions in the translations. In other words, all translators show comparatively strong emotional words in their translation, which may offer us insights into another norm of translating emotions in literary works, namely that translators may often choose high-intensity emotional words either in sharing their joy, strong belief and willingness, or arousing deep sorrow and fear, irrespective of whether they may intend to have their versions generate an immediate yet intense impression on the audience in reading.

Studying the extent to which extent emotions flow differently in the five versions, and what may attribute to the possible differences or similarities, we need anatomize the top 10 high-frequency emotion word lists.

From Appendix I in S1 Appendix, it is evident that in Lin's version, the most frequently used word is "孺子 (ru-zi, child)", a term of appellation in ancient Chinese. It is now rarely used in modern Chinese. It was not listed in the Stop-word List when the emotion words were extracted because this word itself shows the affection of the elders to the younger ones when calling them ru-zi. In Lin's translation, in most cases, ru-zi is what different people call David, of whom most are imbued with love and tenderness, such as Aunt Betsey, Peggotty, and the Doctor. It is true that a few might be with negative feelings, like Mr. and Mrs. Murdstone, but the few times may have little effect on the initial intention underpinning the use of htis word. Therefore, this word substantiates Lin's unique choice of words that as he tries to tell the audience that David is the author's "favorite son" (in the preface of the 1867 edition of *David Copperfield*).

The words, "入时 (ru-shi)" and "时尚 (sh-shang)" which both mean fashionable in modern Chinese are not the same as the words in ancient Chinese. Instead, two characters in the two words both have separate meanings. *Ru shi* means "进入(的)时候 (jin-ru (de) shi-hou), when entering", and *shi shang* are mostly adverbs when appearing together, of which *shi* is an adverb

of time meaning *when* and *shang* may mean *yet*, *too*, *even*, or *still*, etc. Let's look at some examples:

Example 1: 后密考伯入时, 即有公差随入, ……·
Literal translation: Later *when* Micawbr *entered*, a sheriff followed him
Original: …for within five minutes, he returned in the custody of a sheriff's officer, …
Example 2: 至于入时, 微觉为地至狭, 不能展拓。
Literal translation: *When* I *entered (took a seat)*, I felt a slight small space that I could not stretch myself.
Original: …but I certainly could have wished, when we sat down, for a little more room.
Example 3 : (根密支) 至饭时尚哭。
Literal translation: Mrs. Gummidge was still crying at mealtime.
Original: ("I feel it more than other people," said Mrs. Gummidge.) So at dinner.
Example 4: 及归时尚留此状于脑中。
Literal translation: When I got home, it is still kept in my mind.
Original: It haunted me when I got home.

In all likelihood, this reflects the limitation of the dictionary-based emotion analysis when it comes to ancient Chinese. This also presents the development of Chinese in its usage and meaning from ancient to modern written work. Although there are quite a few common features in different forms of one language, this certainly has characteristics of the times. A modern emotion dictionary can only draw words in modern use. Thus a close scrutiny of emotions in Lin's version requires a dictionary of listing emotions in ancient Chinese. Perhaps a better way to find the emotions in Lin's is to list single characters with concrete meanings. A rough attempt at extracting the primary emotions in single characters shows us the most frequently used word is "笑" (*xiao*, smile), 279 times (written as 279 "笑s", similarly hereinafter for other words), followed by 246 "乐s" (*le*, joy), and 194 "怒s" (*nu*s, rage). Besides joy, some typical positive emotions are also common, like 161 "悦s" (*yue*, joy or like/love), 92 "喜s" (*xi*s, happy or like), and 97 "信s" (*xin*, trust/ believe). Except for rage, Lin Shu also adopts many high-intensity negative emotion words, such as 186 "哭s" (*ku*, cry), 139 "悲s" (*bei*, upset), 106 "忧s" (*you*, worry), 93 "忍s (*ren*, tolerate)", 84 "疑s (*yi*, suspect)", 78 "痛s" (*tong*, agony), 69 "惊s" (*jing*, shocked), 58 "恨s" (*hen*, hatred), 55 "惧s" (*ju*, fear) and 53 "伤s" (*shang*, sorrow). Certainly, in some places, these words are modified by valence shifters like negators or adverbs. It is worth noting that some single characters have multiple meanings, like "思 (*si*)" which means missing or thinking (of which the former is an emotion type, but the latter is not). The same is the case for "念 (*nian*)", which can be reading, thinking, missing and worrying. For words of this kind, using Machine-Learning Algorithms might be the best solution.

Overall, the words themselves present strong emotions which are complex and subjective experiences and which may bring physiological or behavioral responses to both internal (from the translator himself) and external (from the original) stimuli [19]. Lin Shu emphasized the great influence of the original, noting that while translating, he could not hold back his emotions but had to let them flow with that of the original. Thus he believed that his emotion behavior was conducted by the author. However, he followed the emotion classification that has been commonly recognized and accepted in Chinese culture, despite physiological data showing variation from person to person. For Lin Shu, emotions are a privilege, and he spares no efforts in translation to achieve a natural flow of emotions, which evidently echoes the laws of emotion flows in China. This may be taken as Lin's norm in translating literature.

In terms of the top ten highly frequent used positive emotion words in modern Chinese versions, Appendix I S1 Appendix shows that "相信" "一定" "希望" "喜欢" "朋友" "高兴" are all chosen as the most frequently used positive word, occupying 3/5 of all top ten. This reflects the fact that modern vernacular Chinese since its birth has remained relatively stable in its use

and meaning, though different translators may have their own preferences or understanding of the original. In Xu's version, "希望" and "相信" are the top two high-frequency emotion words belonging to two subtypes of *Goodness* category, as "希望" is in that of "祝愿 (wish)" and "相信" of "相信 (believe)", with separate valence of +5 and +7. Dong Qiusi chose "相信" and "一定" as the most frequently used emotion words, and the frequencies are much higher than those in the other three translations. The two words belong to the subtype of "相信" with the high valence of +7 and +5. For Zhang Guruo, his top-one frequency positive word is "不错 (quite good)", followed by "一定 (sure/must)" and "高兴". "不错" is comparatively less strong emotional than the other four, with the valence of +3, and it is, in most cases, used as a collo-quial word in speaking. Similarly, Dong and Zhang both have another high-frequency positive word "快活", which is also a common word in speaking. Therefore, there is reason to believe that Zhang has a unique tendency to use colloquial language in his translation, and that he expresses emotions in a much gentler way. Zhuang Yichuan, similar to Xu, adopts "希望" as the most frequent one, and "高兴" is the second. "高兴" in the category of *joy*, is also a word of high intensity valenced +5.

When it comes to high-frequency negative words, despite the fact that word types transpire more than the positive ones, they are not used as frequently as the latter, as is presented in Appendix II S1 Appendix. In general, "死 (death)" "不幸 (unfortunate)" "痛苦 (pain)" "怀疑 (suspect)" "忍受 (tolerate)" "厉害 (fierce)" "消息 (news)" "眼泪 (tears)" and "困难 (difficult)" are all used quite frequently in the four versions. "怀疑"and "忍受" both are the strongest emotions in "恶"category valenced -9, "厉害" in "惧" category is much weaker, with the valence of -3. In the four versions, "死" appears most often in Dong's and Zhang's versions, and "难过" is the top one negative emotion in Xu's and also in Zhang's texts with the same fre-quency as "死". Yet, diverging from the first three translators, Zhuang has "痛苦" the top on the list, followed by "死".

One thing cannot be overlooked in the extraction of emotion words. In Zhang' version, "把 头" appeared 76 times which, as a noun, is an out-of-date word referring to a person in control of a certain place or a trade like porterage in old China, exploiting people or laborers under control, so it is classified as a negative word in the Chinese dictionary. However, in Zhang's version, it is not a word, but a special sentence structure using "把" between a subject and an object, and a verb followed after the object, like "他把头一转 (He turned his head)", meaning a movement of head. Searching this structure in all five versions, we find it only appears in the modern versions, respectively 3, 25 and 62 times in Xu, Dong, and Zhuang. Therefore, this indicates that "把头…" is a special structure in modern Chinese rather than a noun. It might also be the case that the structure was used less at the beginning of the language development. This structure also reflects the inefficiency of natural language processing in extracting words based on a dictionary.

In Martin & White's appraisal system, all emotions can be grouped into 4 sets with 8 sub-sets, respectively, dis/inclination, un/happiness, dis/satisfaction and in/security [20]. Based on the principles of grouping different words into the four sets, those top 10 high-frequency emo-tion words in the four translations are grouped into the 4 sets (see Appendix III S1 Appendix). If the categories in the Chinese dictionary are put together in the same table, it is clear-cut that in Chinese, the mainstream positive emotions are *joy* (PA), *praise* (PH) and *trust* (PG) and negative ones are *sadness* (NB), *fear* (NC) and *worry* (NE). Besides the major ones, some other minor types are also included, like *wishes* (PK), *anxiety* (NI), *apprehension* (NF). All these sub-types are separately distributed into all 8 subsets in M & W's system. Two Chinese words, "朋 友" and "死" are quite special in the table. Though "朋友" in the Chinese dictionary has been grouped into *love/like* (PB), it might be grouped into the latter three sets, since it may reflect one's attitudes, feelings or trusts towards a friend. For "死", it is itself a flexible word (NX),

**Table 6. Comparison of Zhang's emotion words in Martin's system and the Chinese dictionary.**

| Zhang's | | | | | |
|---|---|---|---|---|---|
| | positive | happiness | 喜欢(+5) | 228 | 好 (goodness) |
| | | | 高兴(+5) | 317 | 乐 (joy) |
| | | | 快活(+7) | 163 | |
| | | Security | 相信(+7) | 292 | 好 (goodness) |
| | | | 一定(+5) | 410 | |
| | | satisfaction | 不错(+3) | 417 | 好 (goodness) |
| | | | 神气(+5) | 115 | |
| | | inclination | 希望(+5) | 254 | 好 (goodness) |
| | | | 愿意(+1) | 164 | |
| | negative | unhappiness | 难过(-5) | 141 | 哀 (sadness) |
| | | | 痛苦(-7) | 73 | |
| | | | 不幸(-9) | 66 | |
| | | | 死 (-7) | 141 | |
| | | insecurity | 消息(-1) | 76 | 哀 (sadness) |
| | | | 死 (-7) | 141 | |
| | | dissatisfaction | 苦恼(-5) | 68 | 恶 (disgust) |
| | | | 脾气(-5) | 117 | 怒 (anger) |
| | | | 死 (-7) | 141 | |
| | | disinclination | 害怕(-3) | 83 | 惧 (fear) |
| | | | 厉害(-3) | 78 | |
| | | | 可怕(-3) | 64 | |
| | | | 死 (-7) | 141 | |

**Table 7. Comparison of Zhuang's emotion words in Martin's system and the Chinese dictionary.**

| Zhuang's | | | | | |
|---|---|---|---|---|---|
| | positive | happiness | 喜欢 (+5) | 244 | 好 (goodness) |
| | | | 高兴 (+5) | 375 | 乐 (joy) |
| | | | 愉快 (+5) | 147 | |
| | | security | 相信 (+7) | 191 | 好 (goodness) |
| | | | 一定 (+5) | 351 | |
| | | | 肯定 (+3) | 146 | |
| | | satisfaction | 漂亮 (+6) | 140 | 好 (goodness) |
| | | inclination | 希望 (+5) | 403 | 好 (goodness) |
| | | | 愿意 (+1) | 221 | |
| | negative | unhappiness | 难过 (-5) | 80 | 哀 (sadness) |
| | | | 痛苦 (-7) | 174 | |
| | | | 死 (-7) | 135 | |
| | | insecurity | 怀疑 (-1) | 102 | 恶 (disgust) |
| | | | 惊讶 (-7) | 108 | 惧 (fear) |
| | | | 死 (-7) | 135 | |
| | | dissatisfaction | 卑贱 (-9) | 97 | 恶 (disgust) |
| | | | 情绪 (-5) | 87 | |
| | | | 死 (-7) | 135 | |
| | | disinclination | 害怕 (-3) | 111 | 惧 (fear) |
| | | | 厉害 (-3) | 80 | |
| | | | 可怕 (-3) | 77 | |
| | | | 死 (-7) | 135 | |

which can be used alone or coordinated with other Chinese characters to form a new word. As such, it might be grouped into either of the four negative subsets in different contexts. Comparing the emotion types in both English and Chinese systems, we find emotions do have common features between different languages.

When each version is taken into consideration, as shown in Tables 5–7, it can be seen that most of the emotion words reflect high intensity. In terms of positive ones, Xu and Zhang share a striking similarity in word choice, as only "可爱" in Xu's is replaced by "不错" in Zhang's and the latter reflects one of Zhang's translation features, i.e., he employs quite a lot of colloquial words in his translation. The result again proves the comparative stability of modern Chinese, though different people might adopt spoken language in their work. Dong and Zhuang also enjoyed a certain degree of similarity in choosing words of happiness, security and inclination. Yet they have their own preferences of word choice, like "必须" in Dong's and "肯定" in Zhuang's. Although the two words are both grouped into the security set, they have different degrees of intensity. Zhuang also prefers using "漂亮 (pretty/beautiful)" to "美丽", while the other three use the latter more than the former. Among the four translators, Dong shows the greatest intensity of happiness and security, but focuses less on satisfaction and inclination. On the contrary, the other three have employed "愿意", a word with the weakest intensity to show a sense of inclination. In general, the four translators have shown quite similar intensity of positive emotions in their works, although they have their own uniqueness in choosing emotion words.

For the negative words, it is obvious Xu, Dong and Zhang have chosen the same three words to show unhappiness, yet Zhuang has used just two. Other than that, Xu and Dong also adopt quite similar words to exhibit insecurity, while Zhang and Zhuang share a disinclination to follow a similar path. The four translators choose quite different words for dissatisfaction, in which context Dong again adopts the most intense emotion words in his works.

To sum up, although the four translators have deployed very similar words in expressing emotions, regardless of the time difference, they do have their unique choices when specific emotions are considered. Dong has chosen the words with the strongest emotions, and Zhang tends to use colloquial words. In consideration of the emotion types, they might keep the personal feelings or mental behavioral process towards oneself or the social well-being as the main emotion flow. Particularly, they might depict different characters' inner heart state and mental behavior more than than their activities towards others. Emotions, as a response to meaning, may also be a response to the experienced emotion and its perceived impact on oneself and others [21]. In this respect, modern vernacular Chinese, in general, reflects consistency in expressing emotions over half a century, especially basic ones, like joy, love, sadness, fear and anger.

When Lin's work is included, it can be seen that "一定" "喜欢" "高兴" "痛苦" "怀疑" "忍受" "厉害" and "对不起" do not appear in Lin's works. Instead, Lin employed single characters like "必 (bi, must)" "喜" "悦" "乐" "痛" "疑" "忍" "甚 (shen, fierce/ too much)" "恕 (su, sorry/ pardon)" to express similar or the same meanings of the above words. On the other hand, we also find that Lin Shu adopted two words to translate "must", one is "必", and the other is "坚定 (jian-ding), firm/ determined". For the latter, he changed the emotion subtype "相信 (xiang-xin, believe)" into "赞扬 (zan-yang, praise)".

This result reflects the flexibility of word choice as well as the change in word meanings and usages in different forms of Chinese. From the historical perspective, the Chinese written language has experienced great changes across the span of the ancient to modern periods, and two-character words in modern Chinese can narrow down the emotional scape of one-character words. Thus, emotions expressed in modern Chinese are more concrete and distinct. Further, the result also shows the limitations of using a lexicon, especially a modern Chinese

dictionary to ancient Chinese, to extract emotion words. Many words might also be used as oral Chinese in our daily life at present, but rarely used in written form. Therefore, to analyze emotions in ancient Chinese literature, an ancient-Chinese emotion lexicon needs be developed. This is not an easy task, for one character might have multiple meanings.

## Word type and token respectively in four and five versions

The translations of the same novel undertaken at different times show great similarity, demonstrating that the emotions in Chinese translated works can be analyzed by using the lexicon-based computational techniques. The question arises as to the extent of difference between each text. As has been stated above, quite a few of the words in ancient Chinese are single-character ones, which makes it impossible to extract them by using the modern Chinese emotion lexicon. In such a case, the study will calculate the same word types and tokens in the four and five versions respectively to see how varied they are in emotion expressions (See Figs 2–4).

Fig 3 shows that Dong has the most same word types (47.16% for positive and 41.79% for negative) and tokens (85% for positive and 73% for negative) and Zhang the least word types (33.61% and 29.60%) and tokens (73.12% and 61.00%). This may indicate that they do translate, since their emotion words have so high percentage of similarity. While translating, Zhang has the most unique expression of emotions that diverges considerably from those of others. This corresponds to the fact that he shows a preference for using colloquial expressions in translation in contrast to the other translators. On the contrary, Dong's emotion depictions are more solid. The results also show in general that the four versions share at least one third of emotion types and two thirds of tokens. Surprisingly enough, Xu and Zhuang exhibit very close choices in choosing emotion words. This reinforces the result that Chinese in the modern

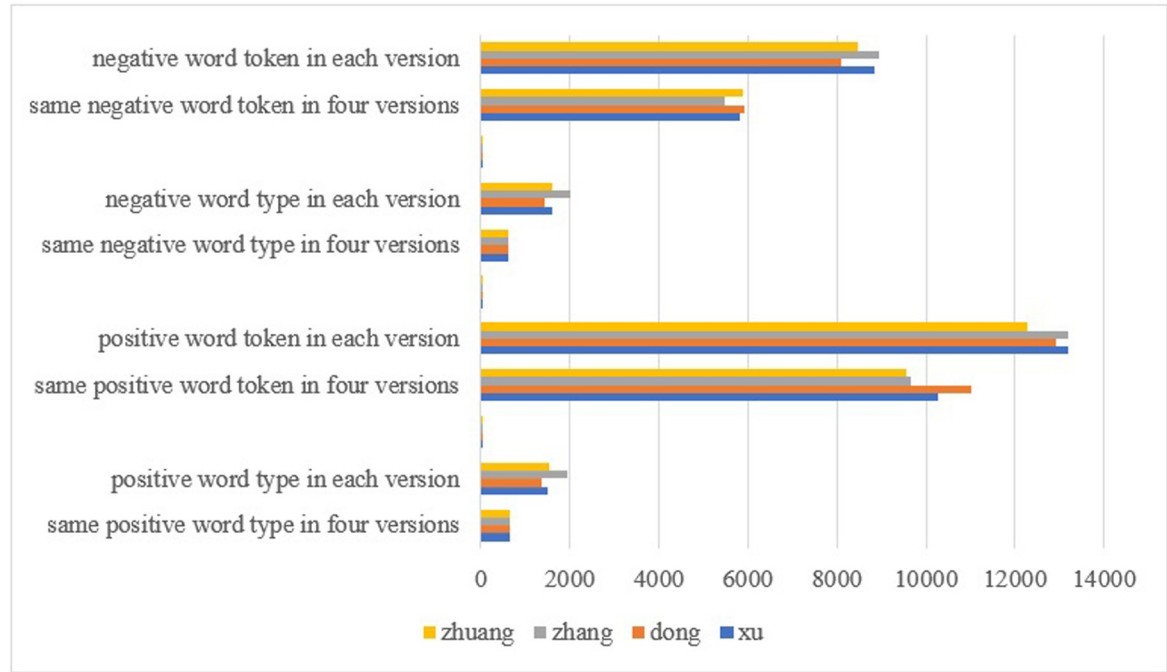

The Same word type and token in four compared with those in each version

**Fig 3. Word type and tokens in different versions.**

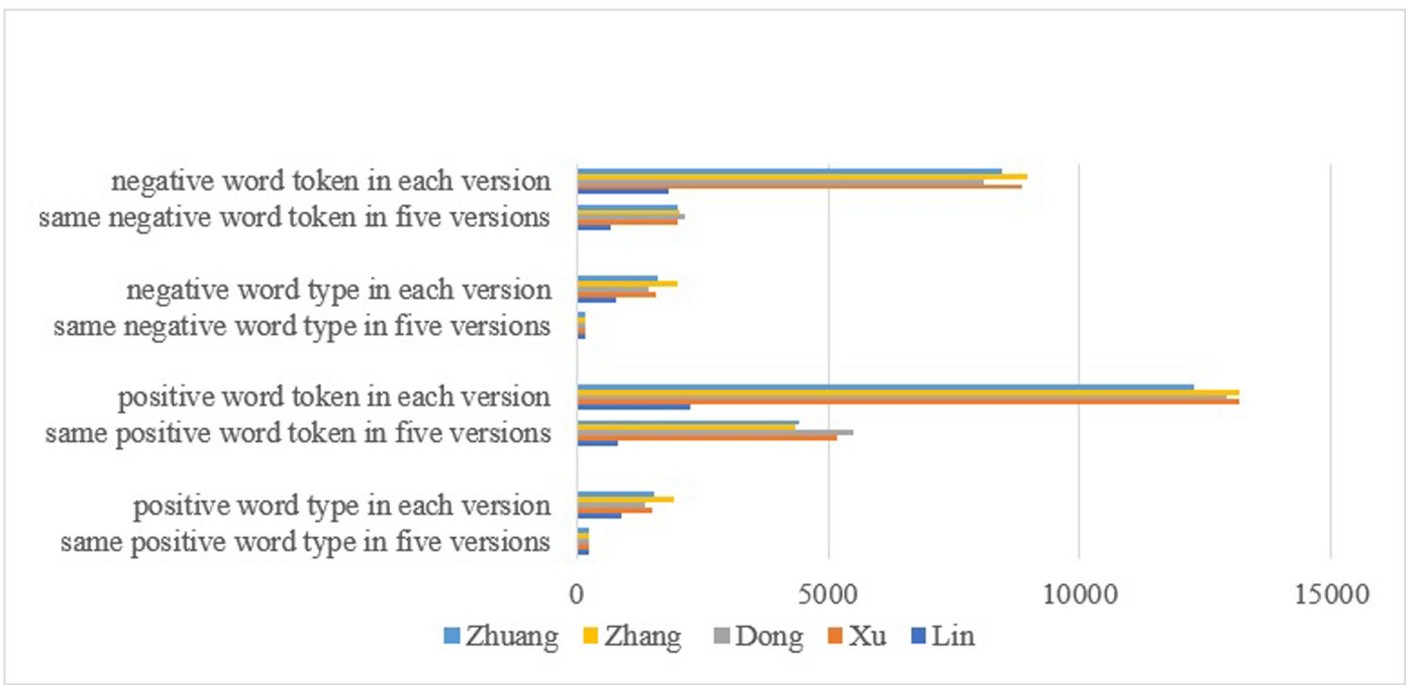

The Same word type and token in five compared with those in each version

**Fig 4. Emotion diversity in translated versions.**

vernacular form has not changed much in its word construction and meaning since its formation.

When the emotions of five versions are concerned, the results vary greatly. Although Lin's emotion words cannot be completely extracted due to the specialty of his language form, he has the most emotion word types (27.38% and 23.46%) and negative word tokens (36.87%) that are the same as the ones chosen by the other four translators. His positive word tokens (36.21%) are close to the percentage shown by Zhuang's (36.18%) but are less than those found in the versions by Xu (39.14%) and Dong (42.76%). On the other hand, Zhang again has the least word types (12.66% and 9.06%) and positive word tokens (32.95%). His negative word tokens (22.72%) are quite close to those found least in the version by Xu (22.69%). According to the data, it is evident that the modern Chinese emotion lexicon has limited applicability in the analysis of ancient Chinese. However, the analysis also shows that quite a few emotion words formed in ancient times enjoy currency in present times as shown in Fig 4.

Combining the above two Figs 3 and 4, we may safely conclude that Zhang has expressed emotions more diversely than others. Dong's emotion expressions are quite flat, whereas Xu and Zhuang enjoy a very similar way of expressing emotions. To a certain degree, Lin's emotional expressions remain in use today.

## Conclusion

This paper analyzed the five translations of *David Copperfield*. Using lexicon-based sentiment and emotion analysis, the three questions framing the study were addressed. In the first instance, the Chinese emotion lexicon was found to work in extracting the emotions in five versions of different times. The result showed that some basic vocabulary in Chinese has maintained stability in concepts and meanings during the evolution of the language from ancient to

modern times [22]. In modern vernacular Chinese versions, the four translations showed substantial similarity in emotion expressions which may reflect the relationship between the original and the translations in aspect of emotions. Reading the same original text, different translators may have experienced different emotional responses. The overall emotion atmosphere may play a decisive role in the translators' judgment towards even localized emotions [23]. In choosing words to express different emotions, translators are influenced not only by the original works but also by the target social context. As social beings, translators' translation behavior accords dynamically with the social and cultural norms in addition to their own subjectivity [18]. Therefore, the expression of emotions by translators in their respective versions is reflective of their personal preferences and social factors. This study also shows the limitation of lexicon-based emotion and sentiment analysis when applied to Chinese translated literature. The lexicon is not expansive enough to include all Chinese words from ancient to modern times, thus restricting the development of a complete picture of how emotions flow in the translations. Future researchers may focus on developing an ancient Chinese emotion lexicon and expanding the modern Chinese one, thus generating more accurate results and giving a much clearer understanding of how each translator has his emotions flow in translations.

## Supporting information

**S1 Appendix. Appendixes I-III.**
(DOCX)

## Author Contributions

**Conceptualization:** Muhammad Afzaal.

**Data curation:** Muhammad Afzaal.

**Methodology:** Yixin Yin.

**Software:** Muhammad Afzaal.

**Writing – original draft:** Yan Li.

**Writing – review & editing:** Muhammad Afzaal.

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
