## [Decision Letter · Decision Letter 0]

27 Dec 2023

A lexicon-based diachronic comparison of emotions and sentiments in literary translation: A case study of five Chinese versions of David Copperfield

PONE-D-23-29324

Dear Dr. Afzaal,

We’re pleased to inform you that your manuscript has been judged scientifically suitable for publication and will be formally accepted for publication once it meets all outstanding technical requirements.

Kind regards,

Luobing Dong

Academic Editor

PLOS ONE

Additional Editor Comments (optional):

Reviewers' comments:

Reviewer's Responses to Questions

**Comments to the Author**

1. Is the manuscript technically sound, and do the data support the conclusions?

Reviewer #1: Yes

Reviewer #2: Yes

2. Has the statistical analysis been performed appropriately and rigorously? 

Reviewer #1: Yes

Reviewer #2: Yes

3. Have the authors made all data underlying the findings in their manuscript fully available?

Reviewer #1: Yes

Reviewer #2: Yes

4. Is the manuscript presented in an intelligible fashion and written in standard English?

Reviewer #1: Yes

Reviewer #2: Yes

5. Review Comments to the Author

Reviewer #1: Dear Author,

I have carefully reviewed your manuscript and commend your efforts in exploring the emotions and sentiments expressed in translations of "David Copperfield" within the Chinese context. This is an intriguing and valuable area of research, and your study adds an important dimension to the understanding of the translation process.

Clarity and Organization: Your manuscript is well-structured and the writing is clear and concise. It effectively introduces the research question and significance of the study.

Methodology: Your use of Lexicon-based Sentiment Analysis (LBSA) is appropriate for analyzing emotions in literary translations. However, it would be beneficial to provide more details about the specific lexicons used and their relevance to the context.

Data Analysis: The explanation of data analysis methods is thorough, and your use of Python and R packages is a robust approach. You've successfully analyzed the emotional content in the five translations.

Conclusions: Your conclusions are supported by the data, and it's evident that translators express unique reactions to the source text. The mention of similarities in the modern vernacular Chinese translations adds depth to the discussion.

Future Directions: Consider discussing potential implications of your findings for the field of translation studies and literary analysis. What could future researchers explore based on your work?

Ethical Considerations: Ensure you address any ethical considerations, particularly when dealing with sensitive content in literary translations.

Figures and Tables: Your figures and tables are informative and well-presented. Continue to ensure their clarity and relevance.

Language and Grammar: Your manuscript is well-edited with minimal grammatical errors. Double-check for any remaining typographical issues.

Citations and References: Make sure to provide complete and accurate citations and references throughout the manuscript. Also, consider expanding the discussion section by referring to relevant literature to further strengthen the context.

Overall, your manuscript makes a significant contribution to the field by conducting a sentiment and emotion analysis of literary translations in Chinese, which is a pioneering effort. I recommend addressing the minor points mentioned above and any other suggestions made by peer reviewers to enhance the manuscript's quality further.

Thank you for your valuable contribution to this area of research, and I look forward to seeing your work progress.

Best regards,

Reviewer #2: it is a pioneering investigation that undertakes a sentiment and emotion analysis of literary translations in Chinese. The study revealed that translators as social beings in the target world express unique reactions towards the same emotion in the original text as well as in literary translations. Yet, the modern vernacular Chinese versions also showcase a similarity in the expression of emotions thus demonstrating the decisive role of the overall flow of emotion in the original plays and in translation. The contribution of this study is significant as it provides insights into how translators’ social and cultural background and emotions towards the texts they are translating influence their linguistic choices.

Overall, I would highly recommend this article to anyone interested in sentiment and emotion analysis of literary translations in Chinese.

6. PLOS authors have the option to publish the peer review history of their article (what does this mean?). If published, this will include your full peer review and any attached files.

Reviewer #1: No

Reviewer #2: **Yes: **Thien Nguyen

---

## [Editor Report · Acceptance letter]

26 Jan 2024

PONE-D-23-29324 

PLOS ONE

Dear Dr. Afzaal, 

I'm pleased to inform you that your manuscript has been deemed suitable for publication in PLOS ONE. Congratulations! Your manuscript is now being handed over to our production team.

Kind regards, 

on behalf of

Dr. Luobing Dong 

Academic Editor

PLOS ONE